

# Investigating the regional contributions to air pollution in Beijing: A dispersion modelling study using CO as a tracer

Marios Panagi[1, 7], Zoë L. Fleming[1*], Paul S. Monks[2], Matthew J. Ashfold[3], Oliver Wild[4], Michael Hollaway[4+], Qiang Zhang[5], Freya A. Squires[6], and Joshua D. Vande Hey[7]

[1]National Centre for Atmospheric Science, Department of Chemistry, University of Leicester, Leicester, UK
[2]Department of Chemistry, University of Leicester, Leicester, UK
[3]School of Environmental and Geographical Sciences, University of Nottingham Malaysia, 43500 Semenyih, Selangor, Malaysia
[4]Lancaster Environment Centre, Lancaster University, UK
[5]Ministry of Education Key Laboratory for Earth System Modeling, Department of Earth System Science, Tsinghua University, Beijing, China
[6]Department of Chemistry, University of York, UK
[7]Department of Physics and Astronomy, Earth Observation Science Group, University of Leicester, Leicester, UK

*Now at Centre for Climate and Resilience Research (CR2), Department of Geophysics, University of Chile, Santiago, Chile
+ Now at Centre for Ecology & Hydrology, Lancaster Environment Centre, Library Avenue, Bailrigg, Lancaster, UK

*Correspondence to*: Marios Panagi (mp558@le.ac.uk) and Joshua D. Vande Hey (jvh7@le.ac.uk)

Abstract. The rapid urbanization and industrialization of Northern China in recent decades has resulted in poor air quality in major cities like Beijing. Transport of air pollution plays a key role in determining the relative influence of local emissions and regional contributions to observed air pollution. In this paper, dispersion modelling (Numerical Atmospheric Modelling Environment, NAME model) is used with emission inventories and *in-situ* ground measurement data to track the pathways of air masses arriving at Beijing. The percentage of time the air masses spent over specific regions on their travel to Beijing is used to assess the effects of regional meteorology on carbon monoxide (CO), a good tracer of anthropogenic emissions. The NAME model is used with the MEIC (Multi-resolution Emission Inventory for China) emission inventories to determine the amount of pollution that is transported to Beijing from the immediate surrounding areas and regions further away. This approach captures the magnitude and variability of CO over Beijing and reveals that CO is strongly driven by transport processes. This study provides a more detailed understanding of relative contributions to air pollution in Beijing under different regional airflow conditions. Approximately 45 % over a 4 year average (2013-2016) of the total CO pollution that affects Beijing is transported from other regions, and about half of this contribution comes from beyond the Hebei and Tianjin regions that immediately surround Beijing. The industrial sector is the dominant emission source from the surrounding regions and contributes over 20 % of the total CO in Beijing. Finally, using $PM_{2.5}$ to determine high pollution days, three pollution classification types of pollution were identified and used to analyse the APHH winter campaign and the 4 year period. The results can inform targeted control measures to be implemented by Beijing and the surrounding provinces to tackle air quality problems that affect Beijing and China.





## 1 Introduction

Beijing has suffered from poor air quality during recent decades owing to rapid urbanization and industrialization. Air pollution can have adverse effects on human health and ecosystems; studies have shown that air pollution is a major contributor to the disease burden in China with over a million attributable premature deaths per year (GBD MAPS Working Group, 2016).

While air pollution is still at very high levels for human health and the environment (Shi et al., 2019), policy measures implemented over recent years have helped to reduce emissions (Jin et al., 2016 and Zheng et al., 2018).

Zheng et al., (2018), have conducted an extensive analysis of reductions in anthropogenic emissions of key species in China. Between 2010 and 2017, it is estimated that China reduced its anthropogenic emissions by 62 % for sulphur dioxide ($SO_2$), 17 % for nitrogen oxides ($NO_x$), 27 % for carbon monoxide (CO), 38 % for $PM_{10}$ (Particulate Matter smaller than 10

µm in diameter), 35 % for $PM_{2.5}$ (Particulate Matter smaller than 2.5 µm in diameter).

A major contributor to air pollution in Beijing is the transportation of pollutants from other regions owing not just to the major regional sources but also to the meteorology and the topography of Beijing (Wang et al. 2018). Beijing is surrounded by mountains to the north and west of the city, with densely populated industrial areas to the south. The arrival of air masses from the south has been observed to correspond to the formation of pollution episodes (haze events) and air from the northwest

leads to the cleansing of the city of pollution (Sun et al., 2016). Zhang et al. (2015) observed the same meteorological pattern with winds from the northwest pushing the pollution away from Beijing, and southerly winds being associated with more polluted days.

Guo et al., 2014 showed that severe pollution episodes in Beijing are largely driven by meteorological conditions. They determined that stagnation usually develops with a weak southerly wind from polluted industrial source regions and that $PM_{2.5}$

concentrations are anti-correlated with the height of the boundary layer (the lower-most layer of the atmosphere governed by turbulent mixing) in Beijing (Miao et al., 2017). Moreover, the southerly winds transporting aerosols into the city in conjunction with low boundary layer (BL) height can produce heavy aerosol pollution events in Beijing. Tie et al., (2015) reported that during a haze event in October 2013, the $PM_{2.5}$ concentration in Beijing was 270 µg m$^{-3}$ in southerly winds. The next day the wind direction changed with winds arriving from the north and the concentration decreased to 50 µg m$^{-3}$. Last but

not least, dust storms in Beijing are resulting from the transportation of natural dust from north-western China, due to dominant north-western winds in the spring (Sugimoto et al., 2003; Liu et al., 2008).

Sun et al., 2013, determined that particulate matter levels are higher in the winter owing to increased use of coal. In a survey in 2012, it was reported that almost 85 % of heating energy and 41 % of cooking energy in rural China came from the use of solid fuels (Tao et al., 2018). The use of solid fuels in the residential sector can affect indoor and outdoor air quality and

can have harmful effects on both human health and climate change (Desai et al., 2004). Wu et al., (2017), determined during a pollution episode in July 2013 that the contribution of aerosol species in Beijing is dominated by transport from outside Beijing, with contributions exceeding 50 % on average.

To implement effective emission controls it is important to understand the role of transport processes that can influence pollutant levels. In this study, we use dispersion modelling to determine the pathways of air masses and with further analysis



to determine the transportation of carbon monoxide (CO) to Beijing. The objective of this study is to investigate the impact of regional meteorology and transport on the air pollution in Beijing, and to understand the influence of regional air pollution on Beijing under differing meteorological conditions.

## 2 Methods

### 2.1 NAME modelling

In this study, we used the UK Met Office's Numerical Atmospheric Modelling Environment (NAME) (Jones, et al., 2007) to track the pathways of air masses arriving at Beijing. A large number of hypothetical inert particles (hereafter referred to as "air masses"), are released and their pathways are tracked backwards in time using meteorological fields from the UK Met Office's Unified Model (Brown et al., 2012). These fields have a horizontal grid resolution of 0.23° longitude by 0.16° latitude and 59 vertical levels up to an approximate height of 30 km. The NAME model was chosen as an appropriate model for this study because it uses high resolution meteorological data of approximately 17 x 17 km, it can predict dispersion over distances ranging from a few kilometres to the whole globe, and it has been used successfully in similar studies looking at the impact of air mass pathways on air quality (e.g. Fleming et al., 2012, Liu et al., 2015, Lowry et al., 2016).

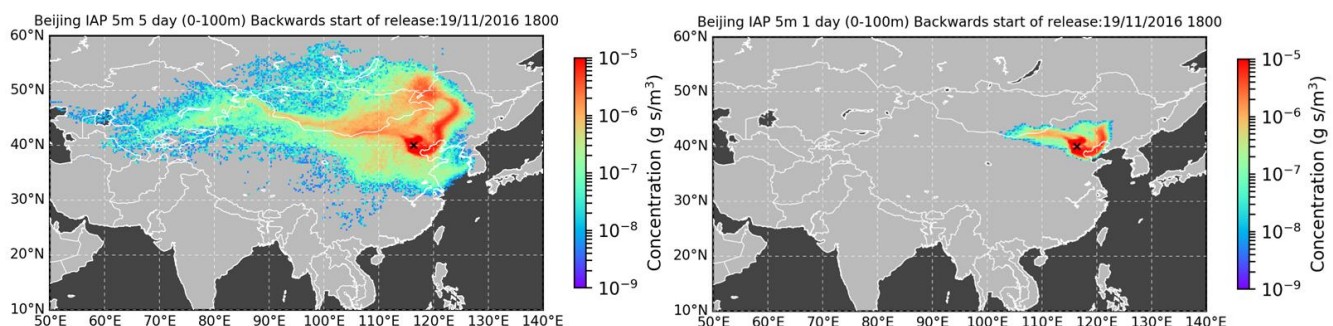

Figure 1: 5 day backwards NAME footprint arriving at IAP meteorological tower (left) and 1 day backwards NAME footprint arriving at IAP meteorological tower (right)

The output is a 3 hourly footprint with a resolution of 0.25° x 0.25°, of all the air masses passing through what we define as the surface layer (0–100 m above ground) during their travel to the specified location. For this study, we modelled 5 day and 1 day backward footprints from 2013 to 2016. Figure 1 shows an example footprint for the 19th of November 2016 at 18:00 UTC (Coordinated Universal Time). The units are based on a release of a known quantity of air masses (hypothetical particles) in grams (g) during an integrated time-period (s) and the results are displayed per grid box, which has a volume component.

To investigate the pathways of air masses arriving at Beijing, we first characterise the air mass pathways based on four quadrants: North-West (N-W), North-East (N-E), South-West (S-W) and South-East (S-E) (see Figure 2). The four regions intersect at the Institute of Atmospheric Physics (IAP) meteorological tower. The choice of the IAP tower as the centre point was made primarily because the modelled meteorological data could be validated against meteorological data measured from the tower. Figure 3 shows that there is a high degree of similarity between the modelled wind speed and direction (from the





meteorological fields) used in NAME and the measured wind speed and direction from the IAP tower. The pathways of the air masses arriving at the IAP tower are important because the chemical and physical composition of an air mass can be influenced by the relative air pollution emissions of the different regions it passes over (Donnelly et al., 2016).

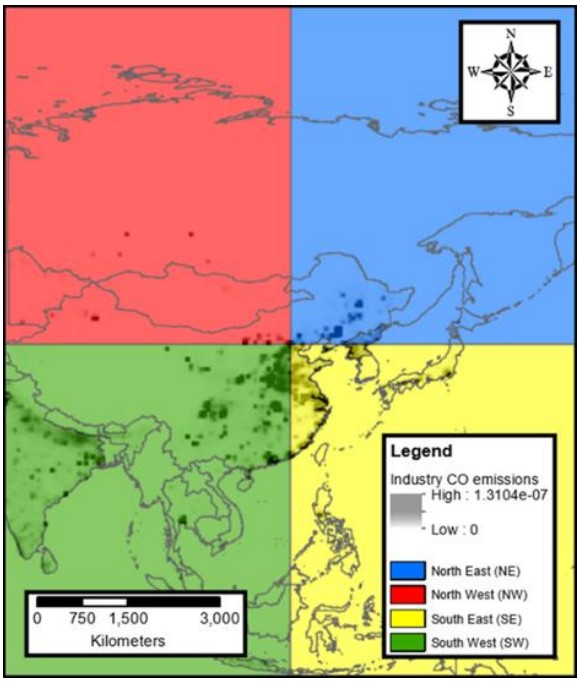

5      Figure 2: The four quadrants created to investigate the air masses arriving at Beijing. Industry CO emissions (kg m$^2$ s$^{-1}$) from MEIC 2013

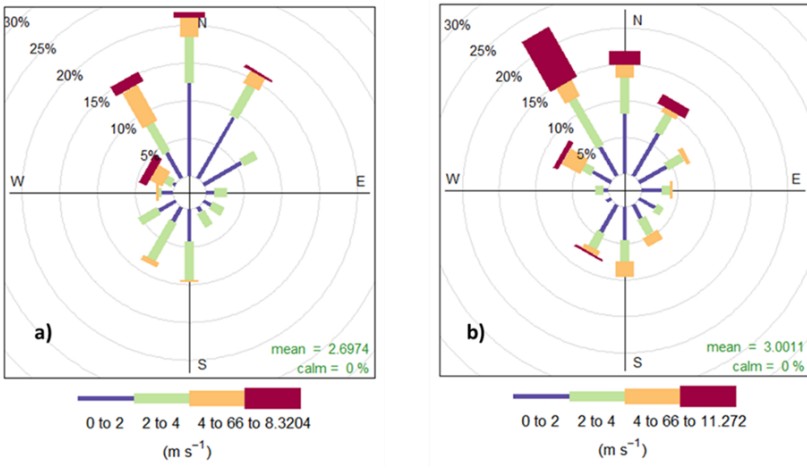

Figure 3: a) Windrose of the measured met data at 100m at the IAP tower, b) windrose of the modelled met data at the IAP tower. Both windroses are for November 2016



## 2.2 CO transport

We combine the derived NAME footprints and the fluxes from the CO emission inventories to calculate a modelled mixing ratio of the emitted species at the measurement site (Oram et al., 2017). We consider only emissions occurring within the timescale of the NAME footprint. CO is commonly used as a tracer of anthropogenic emissions, as its lifetime of 1–2 months

5    and its weak chemical reactivity make it an appropriate tracer for evaluating regional scale atmospheric transport (Naeher et al., 2001. Saide et al., 2011).

In this study, the 2013 monthly Multi-resolution Emission Inventory for China (MEIC, Li et al., 2017) emission inventories were used along with the NAME footprint outputs to calculate the pollution transport to Beijing from surrounding regions. MEIC is a bottom-up emission inventory, which covers 31 provinces in mainland China and includes approximately 700

10   anthropogenic sources. The MEIC emission inventories for CO have a resolution of 0.1° x 0.1° and are available for four sectors: a) the industrial sector, b) the residential sector, c) the transportation sector and d) the energy sector. Details of the inventory development and its source classifications can be found in Liu et al. (2015). The resolution of the emission inventories was reduced to 0.25° x 0.25° to match the resolution of the NAME outputs.

The area of interest was split into "regional" to calculate contributions from outside Beijing and "local" to calculate

15   contributions that are emitted from sources within Beijing (Fig. 4).

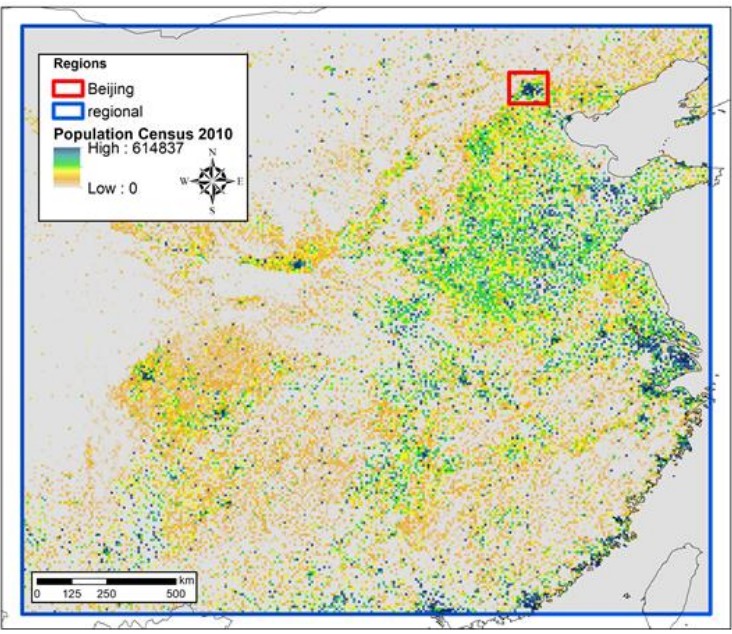

Figure 4: The blue box represents the regional contributions from outside Beijing, the red box is the whole of Local region, and the black box is the Central Beijing and in between is the suburbs region. The map also shows the 2010 population census (people per pixel – WorldPop data)





### 2.3 In situ measurements

In this study, two different measurement datasets were used. The first is CO and $PM_{2.5}$ measurements for 2013-2016 from 12 air quality monitoring stations around Beijing that are part of the national air quality monitoring network run by the China National Environmental Monitoring Centre (CNEMC). The second data set contains CO measurements collected during, the winter campaign of the APHH Beijing project that took place from 5 November to 10 December 2016, at the IAP Meteorological tower (39.975° N, 116.377° E) (Shi et al., 2019). CO measurements at the IAP site were made using six clustered electrochemical sensors (Alphasense Ltd.) encased in a $2 \times 3$ formation. In this study, we use the data collected at the IAP meteorological tower (Shi et al., 2019) to validate the modelled CO data. The CO and $PM_{2.5}$ data from the national air quality network of CNEMC is available on an hourly basis while that from the APHH campaign has a much higher time resolution, 1 Hz. The analysis in this study is intended to aid interpretation of the vast suite of measurements made during the APHH-Beijing campaign, and for setting these campaigns in an annual and inter-annual context.

## 3 Results and Discussion

### 3.1 Analysis of transport to Beijing

The 3 hourly NAME footprints were used to calculate the residence time of the air masses passing over specific regions. The air masses were characterised by season: "Winter" (December, January, February), "Spring" (March, April, May), "Summer" (June, July, August) and "Autumn" (September, October, November), to explore how the pathways change during the year over the four quadrants. The results reveal that the air masses spend more time over the northern quadrants (NW and NE) in all seasons except summer (see Figure 5). In the summer months there is no dominant pattern, with air masses arriving equally from all four quadrants (see Table 1).

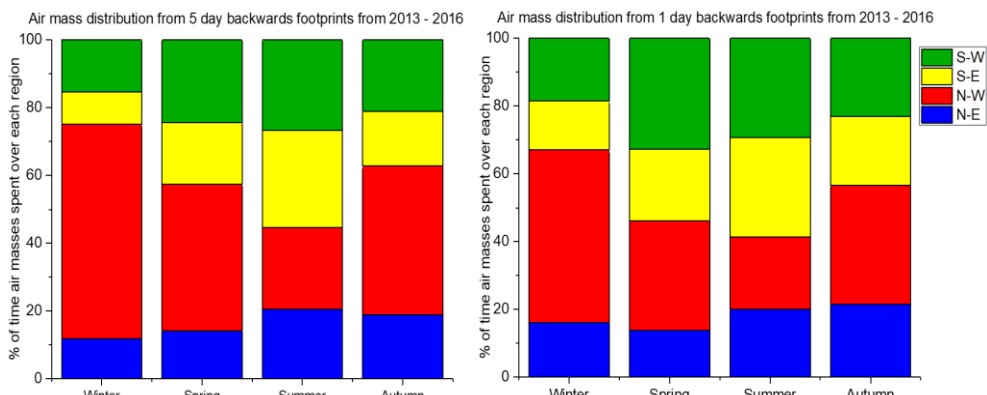

Figure 5: Frequency of the air masses arriving at Beijing from Northwest, Northeast, Southwest and Southeast

To better understand how and where the air masses travel immediately before they arrive at Beijing, 1 day backward NAME footprints were analysed. These footprints reveal the path of the air masses on the last day before they arrive at Beijing. From the comparison of the 1 and 5 day backwards air mass distributions, (Fig. 5) it is observed that the northerly (NW and NE





quadrants) distribution is somewhat lower in the 1 day backward footprints than in the 5 day backward footprints. This indicates that the air masses spend more time over the southern quadrants just before they arrive at Beijing.

Table 1: Air mass frequency (%) over each region for the 5 and 1 day backwards NAME footprints

|  |  | Winter | Spring | Summer | Autumn |
|---|---|---|---|---|---|
| 5 days | **N-E** | 11.9 | 14.2 | 20.6 | 19.0 |
|  | **N-W** | 63.2 | 43.3 | 24.0 | 43.8 |
|  | **S-E** | 9.5 | 18.1 | 28.7 | 16.2 |
|  | **S-W** | 15.4 | 24.4 | 26.7 | 21.0 |
| 1day | **N-E** | 16.1 | 13.8 | 20.0 | 21.5 |
|  | **N-W** | 51.0 | 32.4 | 21.4 | 35.1 |
|  | **S-E** | 14.4 | 21.1 | 29.4 | 20.4 |
|  | **S-W** | 18.5 | 32.7 | 29.2 | 23.0 |

5      It is therefore important to consider both short and long-range air mass pathways. For example, when re-circulation of the air masses occurs, the measured wind speed and wind direction from only one station can be misleading. In Figure 6, two footprints with similar local meteorological conditions are shown; however, the pathways of their arrival at the city are very different, indicating that it is important to have an understanding of synoptic scale meteorology to effectively determine contributions from other areas. Surface meteorological stations in the city alone are not sufficient for working out contributions 10    from regional air mass transport.

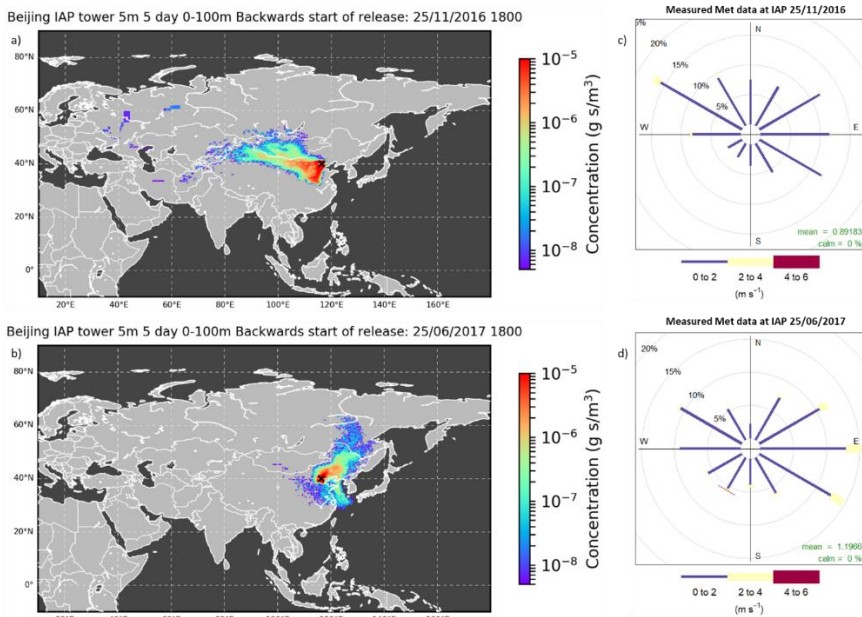

Figure 6: Example footprints of air masses arriving at Beijing from the south (a) and north (b) and wind data measured at the ground during these days (c, d).



## 3.2 Contributions to CO in Beijing

Using the technique described at section 2.2, we calculate the expected CO concentrations arriving at the IAP tower in Beijing at 3 h intervals and compare these with CO measurements from the tower during the APHH campaign (Shi et al, 2019). There is generally good agreement between the measured and modelled CO at the IAP from the 5 day backward trajectories,

with a correlation coefficient of r = 0.756 see Figure 7. Moreover, the correlation coefficient for the 4 year measured CO and the modelled CO in Beijing was calculated, r = 0.692. The approach appears to account well for the meteorologically-driven air mass transport contributions to Beijing pollution and it demonstrates that the variability is governed by the meteorology. Discrepancies between the predictions and observations may be caused by uncertainties in the emission inventory, missing biogenic emissions, secondary CO production, and day to day variability in emissions along with small-scale local sources

which are not included in the emission inventory. Furthermore, there are uncertainties in the meteorological field simulations and owing  to the fact that emissions transported to Beijing from outside the 5 day domain are not included.

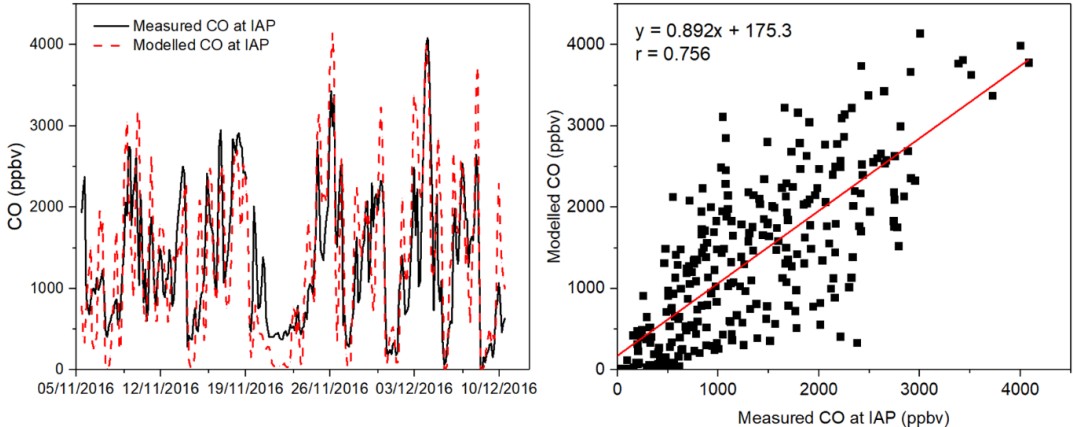

Figure 7: time series of modelled CO vs measured CO at IAP during the APHH measurement campaign period (left), and correlation of measured and modelled CO (right)

Using the CO transportation technique we pair the NAME footprints and the fluxes from the emission inventories on a grid cell by grid cell basis to calculate the CO.  Over the 4 year period from 2013 to 2016 the source sectors contributing most to the total CO pollution modelled in Beijing are the transportation, industrial and residential sectors with approximately 37 %, 34 % and 27 %, respectively. The energy sector is the lowest with a contribution to the total CO in Beijing of less than 2 % in each season.   Sector contributions have different seasonal cycles; The residential sector is highest during the winter months

with an average over 4 years of 637 ppbv (45.3 % of total winter modelled CO) and the industrial sector is highest during the summer with a 4 year average of 355 ppbv (41.3 % of total summer modelled CO). The emissions from the transportation sector are similar throughout each season, however their relative contribution compared to other sectors changes in each season (see sums in Table 2).



By removing local emissions from the MEIC emission inventory, we can isolate the contribution of transport of CO from emissions in other regions. From the 5 day backward footprints, we calculate that over a 4 year period, an average of 45 % of the CO in Beijing was transported from outside Beijing. This high contribution is underscored by the study of Wang et al., (2010) that determined that the contribution of CO from regional sources when local CO emissions in Beijing are controlled by strict control measures was approximately 77 % in Beijing during the 2008 Olympics. During polluted summer days, Chen et al. (2009) reported that approximately 50 % of the CO in Beijing is transported from the Tianjin, Shandong and Hebei regions alone, and Cheng et al., (2018) found that regional transport of pollutants contributed almost 60 % of the $PM_{2.5}$ concentrations in the winter, January 2013, resulting in an extreme regional pollution episode. Furthermore, we determined that the contribution from emissions outside Beijing can reach as high as 80 % of the total CO on some days (as shown in case 2 in section 3.3) .

*Table 2: Sector CO pollution contribution to Beijing from each region*

| Sectors/ Regions | Winter | | Spring | | Summer | | Autumn | | Average | |
|---|---|---|---|---|---|---|---|---|---|---|
| | **ppbv** | **%** | **ppbv** | **%** | **ppbv** | **%** | **ppbv** | **%** | **ppbv** | **%** |
| Industry regional | 192 | 13.6 | 195 | 30.7 | 258 | 30 | 227 | 22.1 | 218 | 24.1 |
| Industry Beijing | 128 | 9 | 61 | 9.6 | 97 | 11.3 | 122 | 11.9 | 102 | 10.4 |
| **Industry Sum** | **320** | **22.6** | **256** | **40.3** | **355** | **41.3** | **349** | **34** | **320** | **34.5** |
| Residential regional | 229 | 16.3 | 70 | 11 | 67 | 7.8 | 89 | 8.7 | 113.7 | 10.9 |
| Residential Beijing | 408 | 29 | 67 | 10.6 | 72 | 8.4 | 166 | 16.1 | 178.2 | 16 |
| **Residential Sum** | **637** | **45.3** | **137** | **21.6** | **139** | **16.2** | **255** | **24.8** | **292** | **26.9** |
| Transportation regional | 52 | 3.7 | 47 | 7.4 | 68 | 7.9 | 61 | 5.9 | 57 | 6.2 |
| Transportation Beijing | 383 | 27.2 | 186 | 29.2 | 282 | 32.9 | 348 | 33.9 | 299.7 | 30.8 |
| **Transportation Sum** | **435** | **30.9** | **233** | **36.6** | **350** | **40.8** | **409** | **39.8** | **356.7** | **37** |
| Energy regional | 7 | 0.5 | 5 | 0.8 | 8 | 0.9 | 7 | 0.7 | 6.7 | 0.7 |
| Energy Beijing | 10 | 0.7 | 4 | 0.7 | 7 | 0.8 | 7 | 0.7 | 6.9 | 0.7 |
| **Energy Sum** | **17** | **1.2** | **9** | **1.5** | **15** | **1.7** | **14** | **1.4** | **13.7** | **1.4** |

Further analysis was conducted to determine the contributions from each emission sector within Beijing and outside of Beijing. The highest contribution from the residential sector for regions outside Beijing is during the cold months of winter (see Table 2 and Figure 8 for seasonal averages for all regions) reaching a maximum 3 hourly contribution of modelled CO concentration of approximately 1200 ppbv. This can be explained by the greater solid fuel burning during these months (Sun et al., 2013). For other seasons (spring, summer and autumn), the industrial sector was the dominant regional contributor to the CO pollution in Beijing, peaking at 1300 ppbv and contributing approximately 20 % of the total CO in Beijing. Furthermore, the changes in concentrations and contributions of each sector each season can be also associated with the changes in the air mass pathways and the increase in emissions from other sectors.





The sectoral contributions within Beijing are very different. The dominant sector throughout the year is the transportation sector, which contributes approximately 30 % of the total CO in Beijing and a maximum 3 hourly concentration of approximately 1600 ppbv. The same increase in the residential sector during the winter months was observed within Beijing as it was observed outside of Beijing.

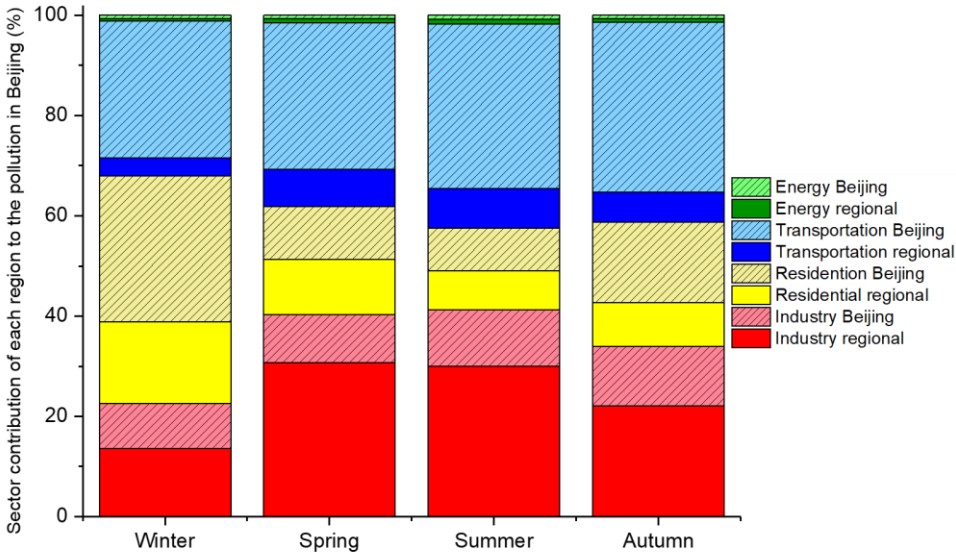

Figure 8: Sector contribution from outside Beijing and within Beijing

The 1 day backward footprints reveal that on average approximately 80 % of the modelled total CO calculated within the 5 days footprints is caused by emissions from within the final 24 hours of air mass transport. Moreover, it was determined that the regional CO air pollution transported from outside Beijing modelled from the 1 day footprints is approximately half of the pollution transported to Beijing is transported from the immediate surrounding regions (such as Hebei, Tianjin etc.) and the other half from regions further away.

From the comparison of where the air masses spend more time before arriving to Beijing with the percentage of the CO air pollution transported from the "Regional China" region (Figure 9), a clear seasonal pattern is observed. The seasonal pattern in the air masses is similar to the the seasonal pattern in the CO contribution from the "Regional China"; when the air masses spend more time over the south quadrants (S-E and S-W) before arriving at Beijing, the regional contribution to the CO pollution in Beijing is higher. This clear seasonality suggests that the regional CO contribution is highly influenced by emissions from the southern regions.

The yearly averages of the modelled CO for 1 and 5 day backward footprints and of the measured CO averaged over the 12 air quality stations in Beijing exhibit no major year to year variations (Table 3). This indicates that since our emissions inventories are the same in each year then the average yearly pathways do not change a lot from year to year.





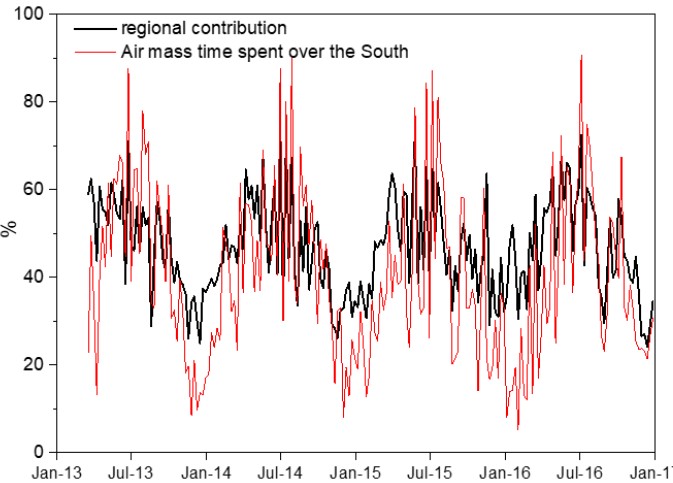

Figure 9: 4 year cycle of contributions from outside Beijing (black line) and the proportion of time (of 5 days) spent over southern quadrants (red line)

Table 3: Yearly averages of the 1 and 5 day modelled CO and the measured average Beijing CO in ppbv

|  | **2013** | **2014** | **2015** | **2016** |
|---|---|---|---|---|
| **Modelled 1 day backwards CO /ppbv** | 721 | 803 | 795 | 786 |
| **Modelled 5 day backwards CO/ ppbv** | 915 | 994 | 989 | 956 |
| **Measured 12 AQMS Average CO/ ppbv** | 1052 | 1005 | 1075 | 962 |

## 3.3 Classification of pollution events

To investigate the relative importance of local and regional emissions during high pollution events from previous years, the data from CNEMC for CO and $PM_{2.5}$ concentrations from 12 monitoring stations in Beijing were used along with the modelled CO. From the measurement dataset, it was calculated that the 4 year average CO concentration is 1024 ppbv and the maximum concentration is greater than 8000 ppbv. The 4 year average $PM_{2.5}$ concentration is 75 µg m$^{-3}$ with a maximum of 606 µg m$^{-3}$. The WHO limit for $PM_{2.5}$ is 25 µg m$^{-3}$ 24 hour mean and the Chinese level 2 limit is 75 µg m$^{-3}$ 24 hour mean (Lin et al., 2018). We also consider $PM_{2.5}$ data here because it is typically the air pollutant of greatest interest from a health perspective (Lelieveld et al., 2015), and is emitted from similar sources as CO such as incomplete combustion from sources like transportation, industry and residential (Staff & Staff, 2002). This is reflected in the correlation between measured CO and $PM_{2.5}$ from all 12 sites used in this study being r = 0.82. $PM_{2.5}$ and CO also have similar trends during pollution events and despite $PM_{2.5}$ having a shorter lifetime than CO, for the short timescales considered here they have similar responses (Shi et al., 2019).

Using the measured CO and $PM_{2.5}$, three classification types have been identified as illustrative of different situations, such as when pollution is dominated by local or regional sources or when these contributions are similar. For these classification types, we defined high pollution events using the $PM_{2.5}$ Chinese limit level 2 (> 75 µg m$^{-3}$, see Figure 10d for the average



PM$_{2.5}$ concentration during each classification) and conducted CO analysis for those periods only. Furthermore, these classification types were also used to analyse pollution events during the winter APHH campaign (see section 3.4).

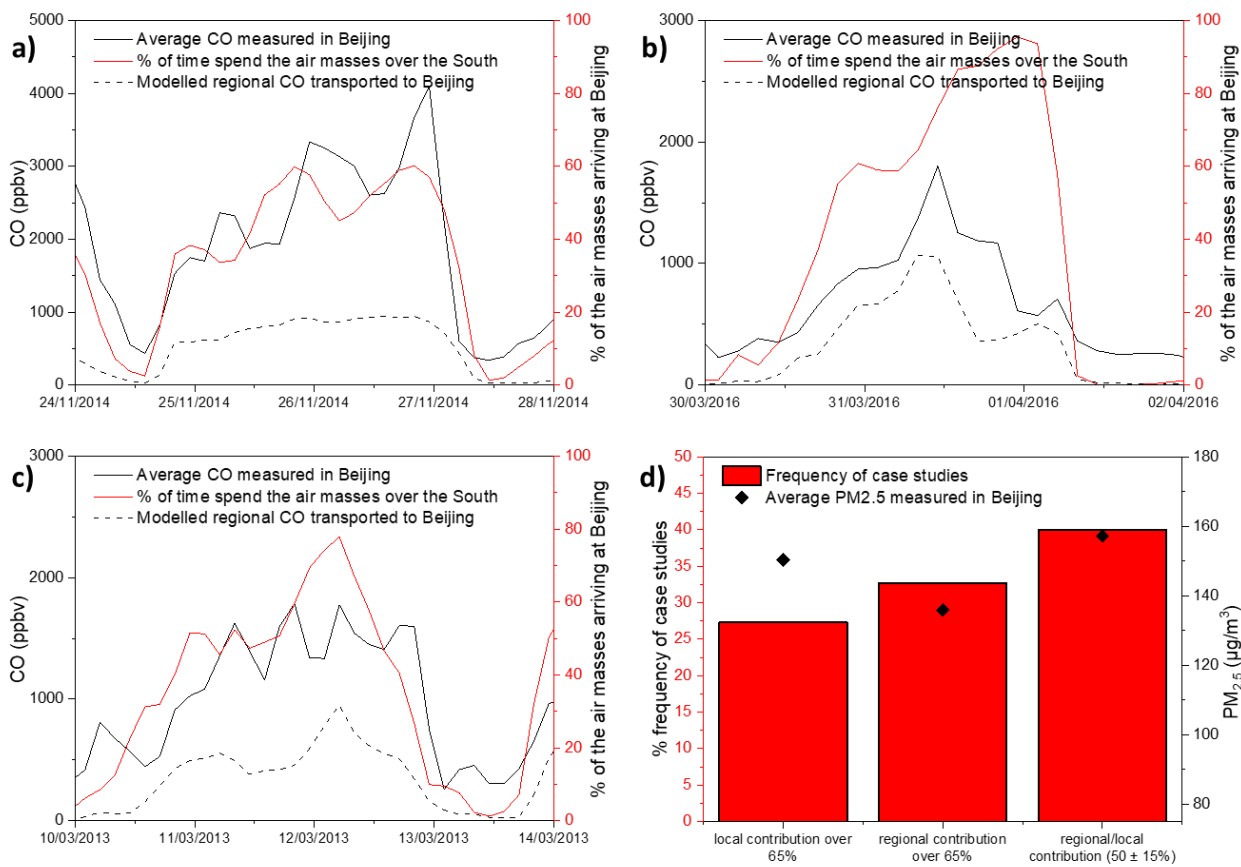

Figure 10: a) b) c) Time series of the measured CO, modelled CO transported to Beijing from other regions and the % of air masses arriving at Beijing from the south for each case study, and d) is the frequency distribution and average PM$_{2.5}$ for each classifications when PM$_{2.5}$ concentration is over 75 μg m$^{-3}$

**Classification type 1: Pollution dominated by local emissions**

Local emissions dominated (defined as > 65 % contribution to) modelled CO pollution in Beijing for approximately 28 % of the time when the PM$_{2.5}$ concentration was over 75 ug m$^{-3}$. Between 24 and 28 November 2014, the pollution levels in Beijing were extremely high with PM$_{2.5}$ and CO concentrations reaching 250 μg m$^{-3}$ and 2500 ppbv, respectively. We find that during this event less than 30 % of CO was transported from outside Beijing (Figure 10a). During these days, the contributions from the local sources were very high, with the transportation and the residential sectors accounting for over 2000 ppbv of the CO in Beijing. The air masses during these days were arriving at Beijing from all directions, which indicates that the transport was possibly very weak.



**Classification type 2: Pollution dominated by regional contributions**

While the $PM_{2.5}$ concentration was over 75 μg m$^{-3}$, approximately 32 % of the time the contribution from emissions outside Beijing dominated (defined as > 65 % contribution to the modelled CO pollution in Beijing). The concentration of $PM_{2.5}$ peaked at over 200 μg m$^{-3}$ on 31 March 2016, accompanied by CO concentrations of around 1500 ppbv (Figure 10b). The air masses arriving at Beijing on this day were dominated by air masses that spent most of their time over the southern quadrants before arriving at Beijing. The modelled regional CO transportation and the dominating southern air mass distribution during this high pollution day suggest that regional transportation from the south was the main driver of the pollution episode in Beijing.

**Classification type 3: similar contribution from regional and local pollution**

When the concentration of $PM_{2.5}$ was over 75 μg m$^{-3}$, the contributions from local and regional sources of the modelled CO was similar (50 ± 15 %) for 40 % of the time during our 4 year study period. An example of these conditions is 10-12 March 2013 when $PM_{2.5}$ and CO concentrations reached 100 μg m$^{-3}$ and 1500 ppbv, respectively. We find that approximately 50 % of the modelled CO was emitted within Beijing and the rest was transported from regions outside Beijing. During the 12[th] of March when the pollution levels were at maximum, the air masses spent more time over the southern quadrants, but the pollution levels decreased rapidly when air masses arrived from the northern quadrants (Figure 10c).

**3.4 APHH Winter Campaign**

During the 35 days of the APHH campaign, 38 % of the modelled CO is arriving from outside Beijing. From the sectors outside of Beijing, the industrial sector is the highest with an average of 180 ppbv and from the sectors within Beijing; the transportation sector is the highest with 424 ppbv during the campaign. Furthermore, the 1 day modelled CO makes up 86 % of the total modelled CO, which is indicating that the majority of the CO transported to Beijing, is found in the immediately surrounding areas. The air masses spend approximately 60 % of the time over the North-west quadrant before arriving at Beijing.

Using the three classifications discussed previously when the $PM_{2.5}$ concentration was over 75 μg m$^{-3}$ during the APHH campaign it was determined that approximately 52 % of the time the pollution was dominated by local contributions, 15 % of the time dominated by regional contributions and 33 % of the time with similar local and regional contributions. The highest $PM_{2.5}$ concentration was recorded when there was a mixture of both local and regional contributions with a 3 hourly average of 400 μg m$^{-3}$.

Furthermore, the average contribution from each classification during the same dates in the three previous years (5[th] of November to 10[th] of December, 2013-2015) were calculated. During the previous three years on average, the pollution was dominated by local contributions on average approximately 51 %, the pollution was dominated by regional contributions 8 % and 41 % of the time the pollution in Beijing was a mixture of similar contributions from both the local and regional sources.





Moreover, during the campaign five haze events were reported (discussed in the APHH project paper, Shi et al., 2019). The haze events are defined as visibility < 10 km at relative humidity (RH) < 80 % and when $PM_{2.5}$ is higher than 75 μg m$^{-3}$ (Li et al., 2019). The classifications above were applied to those pollution events. We found that during the five haze events, the pollution was dominated by local sources and a mixture of both local and regional contributions (see figure 11). Although, the haze events are dominated by local sources, the maximum concentrations are a mixture of both local and regional sources. Two periods between the haze events were polluted but didn't classified as haze events due to the visibility and RH. During the first period on the 13$^{th}$ of November the contributions are a mixture of both local and regional sources and during the second period on the 28$^{th}$ to the 30$^{th}$ of November the pollution was mainly dominated by the local sources.

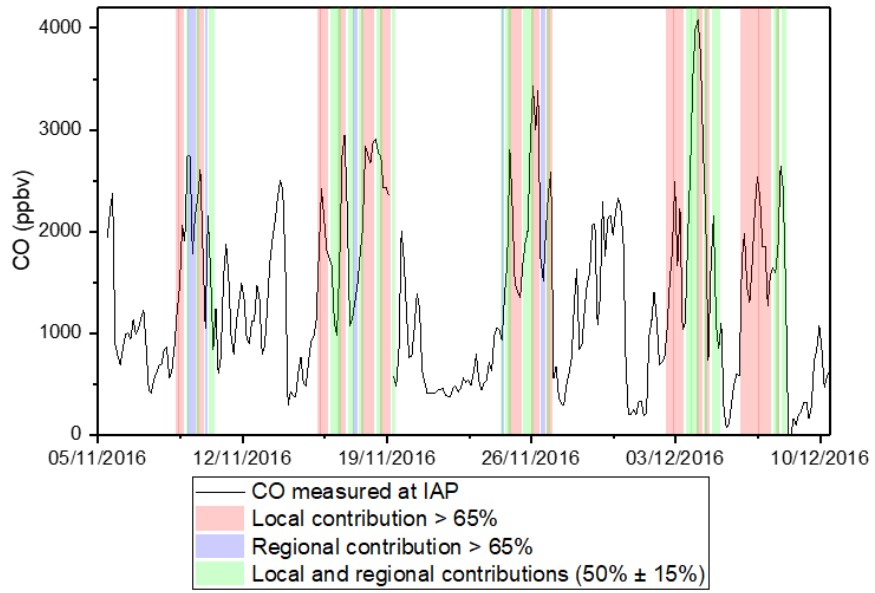

Figure 11: Time series of the measured CO at the IAP during the APHH campaign with the points representing the time where $PM_{2.5}$ concentration was over 75 μg m$^{-3}$. The colours represent the three classifications observed during the five haze episodes identified in Shi et al., 2019

## 4    Conclusions

In this study, we demonstrated that the dispersion modelling approach used provides a valuable way of assessing regional transport impacts and is able to explain well variations in CO over Beijing. The variability in CO is clearly driven by meteorological processes (e.g., good correlation in Fig 6) and reflects varying contributions from local and regional sources.

Averaged over the 4 year period from 2013 to 2016, the highest contributors to the total CO pollution in Beijing are the transportation, industrial and residential sectors with approximately 36 %, 33 % and 29 % respectively. From the results in this paper, it is clear that transportation of pollutants owing to the meteorology in China can strongly influence the pollution in Beijing. The contribution of CO pollution from outside Beijing can account on average over the 4 years for up to


approximately 45 % of the total CO pollution in Beijing, with 23 % being transported to Beijing from the immediate surrounding areas and 22 % from areas further away.

The main contributor to the CO pollution from outside Beijing was the industrial sector with approximately 20 % contribution on average to the total CO in Beijing, resulting from the many heavily industrial areas south of Beijing. The
residential sector contribution from outside Beijing is very high, contributing 45 % of the CO in Beijing during the cold months of the winter, because of the coal combustion and solid fuel burning used for cooking and heating.

Although meteorology and regional transportation of CO are significant contributors to the CO pollution levels in Beijing, the pollution that is emitted within Beijing is still at very high levels with an average of 55 % of CO pollution in Beijing. Within Beijing, the main contributor is the transportation sector with contributions of 25 % of the total CO in Beijing.

The classifications revealed that over the 4 year period, when the $PM_{2.5}$ concentration in Beijing was over 75 $\mu$g m$^{-3}$, 32 % of the time CO in Beijing was strongly dominated by the emissions from outside Beijing and 27 % of the time the pollution was dominated by local emissions. However, the highest mean concentration for $PM_{2.5}$ was observed during the classification where there is similar contribution from source within Beijing and outside Beijing. Furthermore, using the same classifications for the APHH winter campaign 2016, it was observed that 52 % of the time the pollution was dominated by local contributions,
15 % by regional contributions and 33 % from similar local and regional contributions. During the APHH campaign, a slight increase is observed in the regional contributions compared to the previous 3 years.

When the air masses arriving at Beijing spend more time over the southern quadrants increases, the contribution from the emission outside Beijing to the pollution levels in Beijing also increases. This indicates that other pollutants such as $PM_{2.5}$ and Volatile organic compounds (VOCs) could be transported similarly depending upon their lifetime in the atmosphere. Especially
for short-lived pollutants, the results suggested that they can be effectively transported to Beijing from the immediate surrounding regions.

The results lend further weight to the need for a combination of local and regional air quality control measures:

- On average, 55 % of the CO pollution in Beijing is emitted within the city and 45 % is transported in from surrounding regions, emphasising the need for joined up local and regional air quality management strategies.

- Case studies (small sample size) suggest that for approximately 1/3 of air pollution episodes, greater than 65 % of the CO contribution is from regional sources, highlighting the key role transported pollution plays in poor air quality events.

- Control measures may be more effective when they take into account the seasonality both of the regional meteorology and of the relative emissions sector contributions (See Table 2). For example, while Beijing traffic CO
emission reductions should be applied throughout the year, additional controls in the summer might prioritise regional industrial CO reductions, whereas additional winter controls might prioritise Beijing residential CO reductions.

The uncertainties in the datasets (emission inventories, measured data and modelled data), especially in the emission inventories, need to be understood better. Furthermore, more research needs to be done to understand the regional contribution



to Beijing of pollutants such as $PM_{2.5}$, VOCs and ozone. Understanding the effect of the meteorology on the regional transport of pollutants to Beijing pollution in Beijing is very important for new emission control measures.

## Author contributions

MP performed the modelling and numerical data analysis, and led the manuscript development, ZF contributed to the development of the modelling and visualization technique, PSM contributed to research question framing and vision and the manuscript, MA provided CO transport code and ideas on analysis and contributed to the manuscript, OW provided ideas on analysis and contributed to the manuscript, MH converted MEIC inventories for use, ZQ contributed MEIC inventory data, FS collected the CO APHH campaign data, JV oversaw the research and contributed extensively to the manuscript development and data interpretation.

## Competing Interests

The authors declare that they have no conflict of interest.

## Acknowledgments

We would like to thank the UK Met Office for supplying the Unified Model Meteorological data and the use of the NAME model and the CEDA for providing space on the JASMIN supercomputer to run the model. We thank University of Leicester's High Performance Computing services for supplying the necessary computing power for plotting and storing the model output and Duncan Law-Green at Leicester University for developing the code to plot and interpret the NAME model output. We acknowledge James Lee at York University and the national air quality monitoring network run by the China National Environmental Monitoring Centre for providing the data. We would also like to thank the colleagues at IAP for the access to the tower meteorological data. Finally, we would like to thank the National Centre for Atmospheric Science (NCAS) and NERC for funding. JV, OW and MH thank the UK Natural Environment Research Council for support under grants NE/N005406/1, NE/N006925/1 and NE/N006976/1.



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
