# Peer review of "Investigating the regional contributions to air pollution in Beijing: A dispersion modelling study using CO as a tracer"

_Atmospheric Chemistry and Physics, 2019_

## Referee Comment (RC1) · Anonymous Referee #1 · 12 Dec 2019

This manuscript is well written. I recommend it be published with a few minor edits.

Section 2.2: Go into more depth how the footprints and the emissions inventory are combined to obtain atmospheric concentrations of CO.

Figure 4: Black box referred to in caption is not visible.

Page 6, line 10: change the time resolution of 1 Hz to an actual time resolution (in hours, minutes, or seconds).

Section 2.3: State how the CNEMC CO and PM2.5 measurements are used in this study.

[Figure]

Table 2: Add a row for "All sectors regional" and another for "All sectors Beijing"

Page 9, lines 14-15; page 9, line 17; page 10, lines 2-3: I assume the concentrations stated here are from the footprint + EI analysis. Explicitly state this.

Figure 10 caption: change "south for each case study" to "south for three case studies" and include the time frame for 10d.

Page 13, line 20: change "total modelled" to "5 day modelled"

Page 14, line 6: change "but didn't classified" to "but weren't classified"

Figure 11 caption: Don't know what "the points representing the time where PM2.5 concentration was over 75ug/m3" are? Don't see any points.

Page 14, line 15: change "explain well variations" to "explain variations"

Page 15, line 19: change "Volatile" to "volatile"

––––––––––––––––––––––––––––

---

## Referee Comment (RC2) · Anonymous Referee #2 · 24 Dec 2019

This study investigates the regional and local contribution of air pollution in Beijing through a modeling approach. I am generally positive to the topic, presentation, and result. However some issues need to be clarified and justified before potential acceptance.

1. When I read the paper for the first time, I was wondering that is the result of this study a generalized work or a case study? From the conclusions I found the statement being very confirmative and generalizable in Beijing region. If the authors like to make a stronger statement, they should provide a more consolidate proof; otherwise the author should give potential limitation or factors that could be missed in the study, so a

generalization could be difficult to achieve, e.g. the emission inventory is only available at a single year of 2013.

2. The clean air policy in China has been executed starting around 2013, the pm2.5 pollution is seen peaked at 2011-2012 and is largely reduced since. Is this the reason the authors choose 2013-2016 as study period? If not, how much contribution and correlation between the study result and reduction of local emission due to the new policy? If so, 2013 represents a special year, does it mean that we should not over-interpret the result?

3. Does the result from figures 2 and 3 result in the change of area of interest in figure 4? Why the figure 4 is not centered at Beijing? How do the authors define the boundary of regional contribution? (it is difficult to see the "black box" in figure 4). On page 6 the authors discuss the result by quadrants, that is inconsistent with figure 4.

4. When the author uses regression analysis to demonstrate the "account-ability" (P8, l6) of model to the measurement, they should use the "coefficient of determination", instead of the correlation coefficient. The former is the right statistical measure to indicate the proportion of the variance in the measurements that is predictable from the model.

Minors and typos::

P2, l18. Missing parentheses.

P2, l27. Missing parentheses.

P3, l6. Extra comma in citation.

P3, l17 why the resolution is different from l9?

---

## Author Comment (AC1) · 17 Jan 2020

This manuscript is well written. I recommend it be published with a few minor edits.

We thank the reviewer for their positive comments and suggestions. Please find below our replies and the related modifications to the manuscript. The page and line numbers refer to the version of the manuscript published on ACPD.

Section 2.2: Go into more depth how the footprints and the emissions inventory are combined to obtain atmospheric concentrations of CO.

The text below has been added to the section 2.2 (P5, l3):

"To do this, first we derive the sensitivities of the measured air masses to emissions occurring within a grid cell (units $[gm^{-3}] / [gm^{-2}s^{-1}]$, i.e. $sm^{-1}$) and then by multiplying the sensitivities with the emissions from the emission inventories we are able to calculate the modelled CO concentration (dimensionally, $sm^{-1} \times gm^{-2}s^{-1} = gm^{-3}$). To convert the concertation to a volume mixing ratio we divide the modelled concentrations by the molar mass, divide again by the air density and multiply by $1 \times 10^9$."

Figure 4: Black box referred to in caption is not visible.

Figure 4 "black box" – that was a mistake in the caption. Caption for Figure 4 (P5) corrected to say: "Figure 4: The blue box represents the regional contributions from outside Beijing and the red box is the Beijing region. The map also shows the 2010 population census (people per pixel – WorldPop data)"

Page 6, line 10: change the time resolution of 1 Hz to an actual time resolution (in hours, minutes, or seconds).

The time resolution has been changed (P5, l10): to 1 second.

Section 2.3: State how the CNEMC CO and PM2.5 measurements are used in this study.

Text was added at P6, l7 stating how the CNEMC data was used.

"In this study, we use the data collected at the IAP meteorological tower (Shi et al., 2019) to validate the modelled CO data and the CNEMC datasets to compare with our modelled CO from 2013 – 2016 and determine the pollution events in Beijing using the PM2.5 measurements. "

Table 2: Add a row for "All sectors regional" and another for "All sectors Beijing"

Added the 2 rows at the end of Table 2 (P9) as suggested

Page 9, lines 14-15; page 9, line 17; page 10, lines 2-3: I assume the concentrations stated here are from the footprint + EI analysis. Explicitly state this.

Text was added at the beginning of the paragraph at P9, l12 to explicitly say:

"Further analysis was conducted to determine the contributions from each emission sector within Beijing and outside of Beijing using the modelled CO derived from the analysis of 5 day backwards footprints and the emission inventories."

Figure 10 caption: change "south for each case study" to "south for three case studies" and include the time frame for 10d.

Corrected. Caption for Figure 10 (P12) now says: "Figure 10: a) b) c) Time series of the measured CO, modelled CO transported to Beijing from other regions and the % of air masses arriving at Beijing from the south for three

case study, and d) is the frequency distribution and average PM$_{2.5}$ for each classifications when PM$_{2.5}$ concentration is over 75 µg m$^{-3}$ during 2013-2016"

Page 13, line 20: change "total modelled" to "5 day modelled"

Corrected

Page 14, line 6: change "but didn't classified" to "but weren't classified"

Corrected

Figure 11 caption: Don't know what "the points representing the time where PM2.5 concentration was over 75ug/m3" are? Don't see any points.

Corrected. The new caption for Figure 11 (P14) states: "Time series of the measured CO at the IAP in the winter APHH campaign, during the five haze events where PM$_{2.5}$ concentration was over 75 µg m$^{-3}$. The colours represent the three classifications observed during the five haze episodes identified in Shi et al., 2019 "

Page 14, line 15: change "explain well variations" to "explain variations"

Corrected

Page 15, line 19: change "Volatile" to "volatile"

Corrected

---

## Author Comment (AC2) · 17 Jan 2020

This study investigates the regional and local contribution of air pollution in Beijing through a modeling approach. I am generally positive to the topic, presentation, and result. However some issues need to be clarified and justified before potential acceptance.

We thank the reviewer for their comments and suggestions. Please find below our replies and the related modifications to the manuscript. The page and line numbers refer to the version of the manuscript published on ACPD.

1. When I read the paper for the first time, I was wondering that is the result of this study a generalized work or a case study? From the conclusions I found the statement being very confirmative and generalizable in Beijing region. If the authors like to make a stronger statement, they should provide a more consolidate proof; otherwise the author should give potential limitation or factors that could be missed in the study, so a paper generalization could be difficult to achieve, e.g. the emission inventory is only available at a single year of 2013.

It is true that there are limitations due to the limited data and a comment to this effect has been added in the discussion. However we have 4 years (2013 – 2016) of measured CO in Beijing and the yearly sums give as a degree of comfort for the results.

The text below was added to the last paragraph of the section 3.2 in the discussion (P10, l19):

"While our model analysis is limited by the MEIC emission inventory existing only for 2013, the consistent year-to-year CO levels indicate the average yearly pathways do not change a lot from year to year. This indicates that we can confidently use the 2013 MEIC CO emission inventories to understand the importance of meteorological variations in pollution transportation to Beijing. "

2. The clean air policy in China has been executed starting around 2013, the pm2.5 pollution is seen peaked at 2011-2012 and is largely reduced since. Is this the reason the authors choose 2013-2016 as study period? If not, how much contribution and correlation between the study result and reduction of local emission due to the new policy? If so, 2013 represents a special year, does it mean that we should not overinterpret the result?

In this study we are looking at CO contribution and as per our response to reviewer 2 point 1 the CO in Beijing didn't have any dramatic change during the study period. PM2.5 has been decreasing during the years, however here we only use the measured $PM_{2.5}$ as an indicator of pollution events as now mentioned in section 2.3 of the manuscript.

3. Does the result from figures 2 and 3 result in the change of area of interest in figure 4? Why the figure 4 is not centered at Beijing? How do the authors define the boundary of regional contribution? (It is difficult to see the "black box" in figure 4). On page 6 the authors discuss the result by quadrants, that is inconsistent with figure 4.

Figure 4 "black box" – that was a mistake in the caption. Caption for Figure 4 (P5) corrected to say: "Figure 4: The blue box represents the regional contributions from outside Beijing and the red box is the Beijing region. The map also shows the 2010 population census (people per pixel – WorldPop data)"

Figure 2 and 3 are not related. Figure 2 is split into 4 quadrants because we are interested to see where the NAME air masses passed over during their 5 day travel to Beijing. Figure 3 compares the model UM Met data to the measured met data from IAP.

Figure 4 is not centred on Beijing, since Beijing is in the east of China and we are interested to see the contributions from Central and south China to Beijing (since most of the CO sources are found in those regions in China, see Figure 2). The domain was the same between Figure 2 and Figure 4 however, for the regional contributions we are only interested on the sources within China.

4. When the author uses regression analysis to demonstrate the "account-ability" (P8, l6) of model to the measurement, they should use the "coefficient of determination", instead of the correlation coefficient. The former is the right statistical measure to indicate the proportion of the variance in the measurements that is predictable from the model.

Corrected to state $R^2$ (P8, l5,6). The text was changed to say:

"There is generally good agreement between the measured and modelled CO at the IAP from the 5 day backward trajectories, with a coefficient of determination of $R^2$ = 0.571 see Figure 7. Moreover, the coefficient of determination for the 4 year measured CO and the modelled CO in Beijing was calculated, $R^2$ = 0.479."

Minors and typos::

P2, l18. Missing parentheses.

Corrected

P2, l27. Missing parentheses.

Corrected

P3, l6. Extra comma in citation.

Corrected

P3, l17 why the resolution is different from l9?

Forgot to specify that during the study period the meteorological field's resolution changed therefore I added the following in section 2.1:

P3, l9 was added: "These fields from 01/01/2013 to 15/07/2014 have a horizontal grid resolution of 0.35° longitude by 0.23° latitude and from 15/07/2014 to the end of our study period (31/12/2016) have a resolution of 0.23° longitude by 0.16° latitude. Throughout, the meteorological fields have 59 vertical levels up to an approximate height of 30 km."

….

P3, l17 was added: "To avoid inconsistency in the resolution of the outputs due to the two different resolutions of the meteorological fields, all the NAME footprints are outputted with the same resolution."

---

## Author Response (AR2)

**Editor's comments:**

Please include a data availability statement. Refer to https://www.atmospheric-chemistry-and-physics.net/about/data_policy.html

A data availability statement was added to the text that states:

[revised manuscript text omitted]